# Effect of Load Distribution on Trunk Muscle Activity with Lunge Exercises in Amateur Athletes: Cross-Sectional Study

**DOI:** 10.3390/healthcare11060916

**Published:** 2023-03-22

**Authors:** Carlos López-de-Celis, Noé Labata-Lezaun, Sergi Romaní-Sánchez, Sergi Gassó-Villarejo, Erik Garcia-Ribell, Jacobo Rodríguez-Sanz, Albert Pérez-Bellmunt

**Affiliations:** 1Faculty of Medicine and Health Sciences, Universitat International de Catalunya, 08195 Barcelona, Spain; 2ACTIUM Functional Anatomy Group, 08195 Barcelona, Spain; 3Fundació Institut Universitari per a la Recerca a l’Atenció Primària de Salut Jordi Gol i Gurina (IDIAPJGol), 08007 Barcelona, Spain; 4Movement Lab BCN, 08035 Barcelona, Spain

**Keywords:** lunge, electromyography, ipsilateral loading, contralateral loading, bilateral loading, trunk musculature

## Abstract

Background: The effect of load distribution applied to the trunk musculature with lunge exercises has yet to be determined. The aim of this study was to evaluate the effect of load placement using dumbbells on the activation of the latissimus dorsi, erector spinae, external oblique, and rectus abdominis muscles during the lunge. Methods: Forty-two amateur athletes (21 men and 21 women) were recruited. Three lunge exercises were performed with different loading arrangements (ipsilateral, contralateral, and bilateral). The principal variable recorded for muscle activity was mean “root mean square” expressed as the percentage of the maximal voluntary isometric contraction. Results: There are statistically significant differences in the erector spinae (*p* < 0.001; *p* < 0.003) and external oblique muscles (*p* < 0.009; *p* < 0.001) compared with the contralateral side. The muscle on the opposite side of the load achieved higher activation for these muscles. The erector spinae and latissimus dorsi muscle did not reach a statistically significant difference with the contralateral side in any exercise (*p* > 0.05). The higher activation of the latissimus dorsi occurred on the same side on which the load was placed. Conclusions: There was higher activation of the erector spinae, external oblique, and rectus abdominis muscles contralateral to the side of load placement during lunge exercise by amateur athletes.

## 1. Introduction

The use of weights is widely extended to improve muscle activation [1,2,3] and strengthening in subjects with different pathologies [4,5,6] or as general training in the healthy population [7]. Each exercise activates the musculature differently, so knowing this difference in the muscle activation pattern is important. Clinicians must be able to establish a specific and appropriate exercise program that includes correct placement of loads according to the needs of the individual in order to address muscular performance deficits and match the exercise demands with the needs of the individual.

Unilateral exercises are helpful for the rehabilitation of subjects with pathology [8]. These exercises can be more functional for daily activities and more sport-specific than bilateral exercises [9]. Unilateral weight-bearing exercises are commonly prescribed for rehabilitation, and their performance generates the activation of different trunk and lower extremity muscles necessary for the performance of many daily activities [10]. They often involve multiple joints and can be used to improve overall strength, stability, and balance.

One of the exercises in which weight distribution can be modified with dumbbells is the lunge [11,12]. The lunge is a unilateral multi-joint leg exercise that involves knee and hip extensor muscles. A lunge consists of taking a step forward and then lowering the rear knee until it almost touches the ground, reaching a hip and knee flexion close to 90°, and then returning to the starting position. The step distance is approximately 65% of the length of the limb (±5%) [13]. This closed-kinetic-chain exercise is commonly performed in gyms and rehabilitation centers because of its similarity to everyday movement patterns [14,15,16].

The lunge is a popular exercise that can be modified to target different muscle groups and challenge the body in different ways [17]. One of the ways to modify the forward lunge is to change the step distance of the lead lower extremity from the starting position. This can affect the mechanics of the lower extremity by shifting the center of mass and the base of support [18].

Another way to modify the forward lunge is to change the position of the trunk [18]. The rectus abdominis and external oblique muscles play a role in creating intra-abdominal pressure during exertion through the trunk. The rectus abdominis controls lumbar extension, while the external oblique controls lateral flexion and rotation of the trunk [17]. To increase the electromyographic activity of these muscles, exercises such as the Bulgarian squat or standing unilateral dumbbell press can be performed [19]. These exercises involve the use of free weights while standing on one leg, which helps to increase muscle activation in the trunk and lower extremities.

To alter the position of the trunk during a lunge, different weight-bearing position can be used [20]. Thus, the abdominal and back muscles will vary their increased activity to avoid an imbalance of the trunk when performing this exercise.

Dumbbells are versatile and commonly used tools in sports training and can be used to modify the weight-bearing position. They are effective for resistance training and can be used for unilateral and bilateral exercises, which can benefit balance and coordination [21]. Using dumbbells as a part of a sports training program can be a safe and effective way to enhance the training experience [22], and a valuable component of rehabilitative programs for injured athletes and patients [23]. To the best of our knowledge, no study has compared trunk muscle activity depending on the weight-bearing position of the dumbbell while performing a lunge. A tool for monitoring the activity of the muscles involved in an exercise is surface electromyography, which is a widely used technique [24,25] in sports medicine to investigate alterations in muscle activation patterns to improve the development of training and rehabilitation programs [26]. Electromyography (EMG) allows the best exercises to be selected according to the training objective [27,28].

Many studies describe the activation of the hip and lower extremity musculature during a lunge [11,14,18,29,30,31,32,33]. Authors such as Bezerra et al. [33] found differences in EMG amplitudes in the gluteus medius and vastus lateralis that were associated with the dumbbell loading position. The higher activations were observed during FORW (step-forwarding lunge) and WALK lunges (walking lunge). In addition, it has been shown that lunge exercises provide adequate stimulus to the vastus medialis obliquus muscle for strength and endurance training [14]. Some studies have shown altered neuromuscular activity in the trunk musculature in subjects with lower back pain [34,35,36,37,38,39,40]. However, the effect of load distribution applied to the trunk musculature with lunge exercises has yet to be determined.

The aim of the study was to evaluate the effect of load placement using dumbbells on the activation of the latissimus dorsi, erector spinae, external oblique, and rectus abdominis muscles during the lunge.

## 2. Materials and Methods

### 2.1. Study Design

A cross-sectional study was conducted at the Universitat Internacional de Catalunya research laboratory. The local ethics committee of Universitat Internacional de Catalunya–(CEIm Research Ethics Committee) approved the study protocol (study Code: *FIS-2022-09*). The study procedures were conducted following the Declaration of Helsinki (64th World Medical Association General Assembly Fortaleza, Brazil, October 2013). Informed consent was obtained from all participants.

### 2.2. Sample Size Calculation

The sample size calculation was performed using G*Power v.3.1.9.7. The calculation was based on an eta-squared mean effect size (ŋ^2^) of 0.06, similar to that obtained in the study by Stastny et al. [11] in their study of changes in muscle activity in the lower extremity. An ANOVA test was used: “Repeated measured, within-between interaction”. Assuming a bilateral contrast, a significance level (α-error) of 0.05, a statistical power of 0.05 (β-error), and an effect size f of 0.25 were used. The sample size was 42 volunteers.

### 2.3. Sample

Forty-two amateur athletes from the Faculty of Medicine and Health Sciences of the Universitat Internacional de Catalunya were recruited between October 2022 and December 2022. Participants were informed verbally and in writing about the study procedure, and if they consented to participate, they signed an informed consent form.

The study population consisted of amateur athletes over 18 years of age who regularly engaged in fitness activities (at least two days a week). Volunteers were excluded if (1) they had suffered an injury during the last two months or were unable to perform the required physical activity; (2) they did not understand the information provided by the evaluator or the patient information sheet; or (3) they were following pharmacological, medical treatment that could interfere with the measurements, such as treatment with anticonvulsants or antidepressants.

The demographic characteristics of the sample are summarized in Table 1.

### 2.4. Outcomes

EMG recording was performed with the mDurance^®^ device (mDurance^®^ Solutions SL, Granada, Spain). The mDurance^®^ system consisted of an EMG Shimmer3 unit (Realtime TechnologiesLtD, Dublin, Ireland). The unit was a bipolar surface electromyography bipolar sensor for the acquisition of muscle activity. Each Shimmer3 had two channels, with a sampling rate of 1024 Hz applying a bandwidth of 8.4 Hz, and a 24-bit signal with an overall amplification of 100 to10,000 *v/v* [41]. This device is valid and reliable for recording muscle activity during a functional task (ICC = 0.916; 95% CI = 0.831−0.958) [41].

On the other hand, the mDurance Android app received the data from the Shimmer3 and sent it to a cloud service [41] where the data were stored, filtered, and analyzed [41].

For processing and filtering of the raw data, both isometric and dynamic tests were filtered using a fourth order Butterworth band pass filter with a cut-off frequency of 20–450 Hz. The signal was smoothed using a window size of 0.025 s root mean square (RMS) and an overlapping of 0.0125 s between windows [41]. The maximal voluntary isometric contraction (MVIC) was calculated using the peak of the RMS signal during an isometric test, which is carefully explained below. The principal variable recorded for muscle activity was mean “root mean square” (RMS), expressed as a percentage of the maximal voluntary isometric contraction (%MVIC).

The surface electrodes used were 5 cm self-adhesive Valutrode^®^. They were placed on the muscle belly according to SENIAM project recommendations [42] and with an interelectrode distance of 20 mm [41]. The procedure for electrode placement and collection is explained below.

### 2.5. Procedure

Initially, the descriptive variables of the sample (age, sex, hours of seminal sports activity, type of sport, and dominance of the lower extremity) and the anthropometric variables (height, weight, body mass index, and length of the lower extremity) were collected. The measure of the lower limb length was taken with the subject standing, from the greater trochanter to the ground.

#### 2.5.1. Preparatory Procedures

The length of the lunge was also calculated. The procedure of Bouillon et al. [13] was followed, establishing 65% (±5%) of the length of the lower limb.

The distance on the ground corresponding to 65% of the length of the lower extremity was marked. In addition, tape was placed on the floor marking the starting and final positions where the subject had to reach with the heel of the foot when performing the front lunge.

The subject was explained the correct way to perform the lunge. The exercise consisted of a step forward with the corresponding leg according to the previous randomization, until reaching the ground mark with the heel. Then, hip and knee flexion close to 90° without touching the ground with the rear leg knee was performed.

A soft cushion of about 10 cm was placed on the floor to standardize the lowering amplitude. After touching the cushion with the knee, volunteers were instructed to return to the starting position with full knee and hip extension of both legs.

The adjustable weight dumbbells were previously prepared with the weight with which the volunteers should perform the exercise. The weight was 15% of the subject’s body weight. In the bilateral loading exercise, the weight was distributed between two dumbbells, one for each side.

#### 2.5.2. Electrode Placement

This study measured the muscle activation, with electromyography, of four trunk muscles bilaterally (external oblique, rectus abdominis, latissimus dorsi, and erector spinae) during three lunge exercises.

Volunteers were allowed to warm up by performing five repetitions of each exercise required in the study.

After the warm-up, the surface electrodes were placed on the target muscles. The 5 cm self-adhesive Valutrode^®^ surface electrodes were placed following the recommendations of the SENIAM project, ref. [43] with an interelectrode distance of 20 mm [44].

In the latissimus dorsi muscle, the electrodes were placed with the patient upright and the arms hanging vertically. The electrodes were placed obliquely, at about 25° to horizontal, in an inferomedial direction and 4 cm below the angle of the scapula [45].

For the spinal erectors, the electrodes were placed vertically 3 cm from the spine, at the level of the L3 vertebra. The ground electrode was placed on the posteroinferior iliac spine [45].

In the external oblique muscle, the electrodes were placed obliquely approximately 45° parallel to the line joining the lowest point of the costal margin of the ribs and the contra lateral pubic tubercle, above the anterosuperior iliac spine at the level of the umbilicus [45]. The neutral electrode was placed over the anterosuperior iliac spine.

In the rectus abdominis muscle, the electrodes were positioned vertically and centered on the muscle belly at the midpoint between the umbilicus and the xiphoid process (3 cm from the midline) [45].

#### 2.5.3. Normalisation of the EMG

The maximum voluntary isometric contraction test was performed with the electrodes in place to normalize the electromyography data.

The electrography signal was measured with the mDurance^®^ device. The latissimus dorsi muscle was assessed with the subject in a prone position with the shoulder abducted 0° and extended to the maximum active range of movement tolerated by the subject, with resistance in the distal third of the arm in the direction of shoulder flexion [45].

As for the erector spinal muscles, the subject was placed in prone decubitus with the trunk fully extended, requesting an extension to the maximum range of active movement tolerated by the subject, with the hands clasped behind the head. Resistance was exerted in the interscapular region [45].

For the external obliques, the subject was in supine decubitus with hips and knees flexed at 90°. Feet supported, and trunk flexed to the maximum active range of motion tolerated by the subject (curl-up position) and rotated contralaterally (i.e., to the left to assess the right obliques and vice versa). Resistance was exerted on the anterior part of the shoulders towards the flexion and rotation of the trunk [45].

In the case of the rectus abdominis, with the subject in supine position with the hips and knees flexed to 90°, feet supported and the trunk flexed to the maximum active range of movement tolerated by the subject (curl-up position). Resistance was exerted externally towards trunk extension [45].

Once the data for the normalization of EMG were obtained, the lunge was performed with different load placements (unilateral, contralateral, and bilateral). The order of the exercises, as well as the front leg, were previously randomized. For the randomization process, a randomization list was generated before the athletes were recruited using a computer program (www.random.org) that generated a list of random numbers. Subjects performed all three exercise modalities and thus ended their participation in the study (Figure 1).

#### 2.5.4. Lunge

All exercises began with the subjects standing with their feet together at the starting mark and their limbs parallel to the trunk. The dumbbell was held on the same side as the initiating leg in the ipsilateral version, on the opposite side in the contralateral version, and on both sides in the bilateral one (Figure 2). The step was performed up to the mark previously placed. In this position, five descents were performed at a rhythm of 2 s down and 1 s up. For each descent, the subject had to keep the trunk upright and descend until the back knee touched the cushion placed on the floor. In this way, the height of the exercises was standardized. The verbal command was “lunge down as far as possible” [31]. At the end of the five repetitions, the subject returned to the initial position, ending the registration. One minute of rest was given between each lunge exercise.

### 2.6. Statistical Analysis

For the statistical analysis, IBM SPSS Statistic version 20.0 (Armonk, NY, USA: IBM Corp) software was used. Descriptive analysis was carried out. The mean and standard deviation were calculated for quantitative variables. Frequencies were calculated for demographic qualitative variables. The Shapiro–Wilk test was used to determine non-normal distribution of the quantitative data.

A linear mixed model 2 × 3 (multivariate) was performed. A fixed between-subject factor (anterior−posterior) and repeated measures fixed factor (type of exercise) were used for each dependent variable RMS (rectus abdominis, external oblique, spinal, and latissimus dorsi). If the assumption of sphericity was violated, the Greenhouse−Geisser correction was used for interpretation. When a statistically significant effect was observed, a post hoc analysis was performed and the Bonferroni correction was used to adjust for multiple comparisons. These ANOVA models used regression models, so the covariance was constant. This statistic was also repeated with the “sex” covariate.

Effect sizes were calculated using eta squared (ŋ^2^). An effect size > 0.14 was considered as large, around 0.06 was medium, and <0.01 was small. The significance level was *p* < 0.05.

## 3. Results

In the latissimus dorsi, there was no significant interaction between the lunge type and loading side for the women (F = 1.955; *p* = 0.155) or men (1.711; *p* = 0.171).

In the erector spinae, there was a significant interaction between the lunge type and loading side for women (F = 10.089; *p* = 0.0041) and men (F = 9.005; *p* = 0.002). Bonferroni post hoc tests revealed significant differences in the percentage of RMS between the two sides of the same muscle depending on the loading side, both in ipsilateral loading and contralateral loading, but not in the bilaterally loaded lunge (Table 2).

For the rectus abdominis, there was no significant interaction between the lunge type and loading side for women (F = 1.278; *p* = 0.290) or men (F = 3.928; *p* = 0.053).

In the external oblique, there was a significant interaction between the lunge type and loading side for women (F = 7.110; *p* < 0.012) and for men (F = 14.817; *p* < 0.001). Bonferroni post hoc tests revealed significant differences in the percentage of RMS in contralateral lunge between the anterior and posterior side (Table 2).

Table 3 provides information on the differences between the types of lunge according to the side before or after loading, differentiated by sex.

## 4. Discussion

This study aimed to evaluate the effect of load placement using dumbbells on activating the latissimus dorsi, lumbar spine, external oblique, and rectus abdominis muscles during the lunge stride. The tested subjects repeated a lunge three times, each time with a different load placement, evaluating the electromyographic activity of four trunk muscles. Greater electromyographic activity of the erector spinae, external oblique, and rectus abdominis muscles was observed on the side opposite to that of the load. However, no statistically significant differences were found for the rectus abdominis. On the other hand, the most significant electromyographic activity for the latissimus dorsi muscle occurred on the side of the load, except for the lunge ipsilateral load in men. However, no statistically significant differences were reached in any of them.

Although no statistically significant differences were reached between males and females, the activation pattern was generally higher in females. Differences in force generation and activation pattern of the upper extremity during lifting tasks have been described independently of the lifted mass [46].

Bolglia et al. [47] conducted a study to evaluate electromyographic activity in the trunk, hip, and knee muscles during unilateral weight-bearing exercises. They found a similar pattern to ours, where they commented that the greater activation of the external oblique and lumbar spinae might be because the subjects kept the trunk vertical during the exercise. This greater electromyographic activity also occurred in the lunge, requiring greater control in the frontal plane. It also suggests that greater activation of the external oblique and lumbar spine occurred when performing a lateral contraction of the trunk, which in our case counteracted the effect of the load. Counteracting the effect of loading with the activation of muscles that are at a greater distance was also observed in the studies of Graber et al. [10] and Stastny et al. [11], which agrees with the Lauder et al. [48] study comparing unilateral and bilateral dumbbell lifting. They found an asymmetry in activation, which was more significant on the opposite side of the load in the case of unilateral lifting in the lifting phase. Arokoski et al. found that simple therapeutic exercises effectively activated abdominal and erector muscles. By modifying limb and trunk positions or unbalancing trunk movements, it was possible to increase trunk muscle activities [49].

The lunge can be a helpful exercise for rehabilitating subjects with low back pain. Dumbbells seem suitable for beginners performing the lunge exercise due to the low joint impact and increased stability [12]. In addition, the muscle activity generated in our study was between 6.9 and 37.2% MVIC. It could be a helpful exercise for patients with lumbar pathology.

Subjects with low back pain have an altered activation of the trunk musculature [34,35,36,37,38,39,40]. With these subjects, we could place the load on the same side of the pain in the initial phases and then shift the load to the contralateral side for greater activation. Our study used 15% of the subject’s body weight to standardize the load. A heavier load has been seen to increase the stability of the lumbar spine producing more significant trunk activity [34]. Increased loading could be another factor for progression in rehabilitating these patients. Bezerra et al. [33] studied the activation of three lunge modalities with bilateral loading with a difference in execution (static lunge, step-forwarding lunge, and walking lunge). They observed a lower electromyographic activity of the lumbar muscles in the step-forwarding lunge, which was the modality used in our study. The walking lunge version had the highest electromyographic activity. This lunge variation could be used as a lower-activity exercise to progress to higher muscle activity modalities gradually. Calatayud et al. [50] found high activation patterns in a water-bag jerk lift in the erector spinae and external oblique muscles, which were higher than those found in our study with bilateral load; therefore, this exercise could be considered a more advanced modality within the progression.

Conducting studies on subjects with pathology will be necessary to assess whether activation follows these patterns and to determine the necessary load percentage to achieve better activation.

This study has some limitations. Lunge distance was normalized with leg length and squat depth to account for individual variation in the clinical setting. However, there was no control for correct trunk alignment or other factors, such as variability of the sporting gesture, so future studies should control for these factors. In addition, although the subjects were athletes, there could have been a variation in training level and sport load.

## 5. Conclusions

Higher electromyographic activity of the erector spine, external oblique, and rectus abdominis muscles contralateral to the load placement has been observed during lunge performance in amateur athletes. In comparison, the latissimus dorsi musculature is activated more on the side where the load is located.

The dumbbell lunge exercise may be helpful for rehabilitating subjects with altered lumbar muscle activation patterns. The progression of load placement from the side of the lower activation to the contralateral side may be a safe progression that could be increased with a progressive increase in dumbbell weight.

## Figures and Tables

**Figure 1 healthcare-11-00916-f001:**
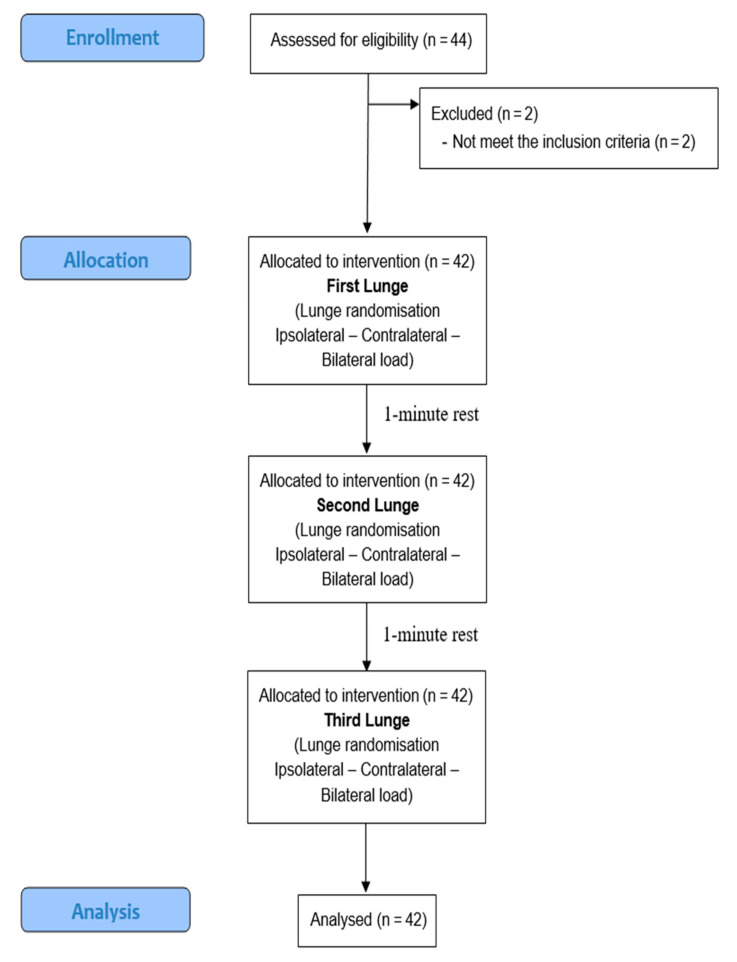
Consort Flow Chart.

**Figure 2 healthcare-11-00916-f002:**
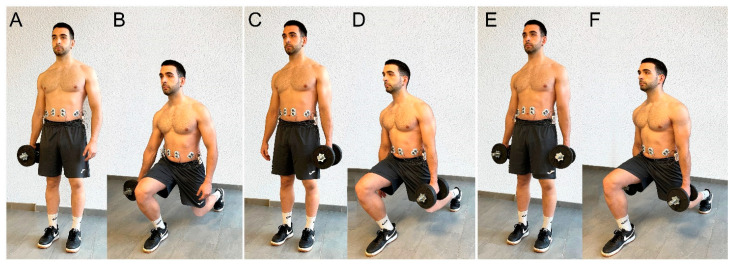
(**A**,**B**), Lunge with an ipsilateral load; (**C**,**D**) lunge with a contralateral load; (**E**,**F**) lunge with bilateral load.

**Table 1 healthcare-11-00916-t001:** Subjects’ demographic characteristics.

	Men (n = 21)	Women (n = 21)
Age (years)	23.71 ± 4.71	24.33 ± 9.43
Height (meters)	1.78 ± 0.04	1.67 ± 0.06
Weight (kilograms)	72.52 ± 8.68	57.29 ± 8.56
BMI (kg/m^2^)	22.87 ± 2.64	20.56 ± 2.69
Dominance		
Right	15 (71.4%)	19 (90.5%)
Left	6 (28.6%)	2 (9.5%)
Type of Sport		
Fitness	12 (57.1%)	10 (47.6%)
Track sport	3 (14.3%)	3 (14.3%)
Field Sport	1 (4.8%)	3 (14.3%)
Others	5 (23.8%)	5 (23.8%)
Weekly hours of sport	6.71 ± 3.54	5.98 ± 3.88

Abbreviations: BMI, body mass index.

**Table 2 healthcare-11-00916-t002:** Descriptive results and differences for sex between sides according to the type of lunge load on the different muscles studied.

	Anterior Side	Posterior Side	Difference between Anterior–Posterior Side			
	RMS (%MVIC)	RMS (%MVIC)	Mean	95% CI	*p*	ŋ^2^
MEN						
Lunge Ipsilateral Load						
Latissimus dorsi	10.00 ± 7.37	11.02 ± 9.74	−1.02	−5.73; 3.69	0.655	0.00
Erector spinae	9.87 ± 4.69	16.24 ± 8.42	−6.37	−10.65; −2.09	0.006	0.18
Rectus abdominis	6.32 ± 5.12	8.44 ± 8.56	−2.12	−6.61; 2.37	0.336	0.02
External oblique	17.69 ± 13.07	26.83 ± 17.81	−9.14	−19.43; 1.16	0.079	0.08
Lunge Contralateral Load						
Latissimus dorsi	8.69 ± 4.50	14.00 ± 14.53	−5.31	−11.44; 0.83	0.086	0.06
Erector spinae	15.05 ± 13.07	10.36 ± 6.20	4.69	0.15; 9.23	0.044	0.05
Rectus abdominis	6.86 ± 3.95	5.35 ± 3.78	1.51	−0.62; 3.64	0.155	0.04
External oblique	35.16 ± 26.44	15.36 ± 9.90	19.80	7.61; 31.99	0.003	0.20
Lunge Bilateral Load						
Latissimus dorsi	8.12 ± 4.38	10.16 ± 8.29	−2.04	−1.31; 5.40	0.219	0.02
Erector spinae	11.40 ± 4.94	14.07 ± 10.99	−2.68	−7.66; 2.31	0.277	0.02
Rectus abdominis	5.39 ± 3.48	5.37 ± 3.52	0.17	−2.01; 2.05	0.987	0.00
External oblique	18.39 ± 13.27	15.51 ± 9.28	2.89	−1.59; 7.36	0.193	0.01
WOMEN						
Lunge Ipsilateral Load						
Latissimus dorsi	23.48 ± 26.96	17.03 ± 9.98	6.46	−5.40; 18.31	0.269	0.02
Erector spinae	11.51 ± 7.22	23.89 ± 19.51	−12.38	−18.79; −5.96	0.001	0.15
Rectus abdominis	7.57 ± 4.91	10.01 ± 5.75	−2.43	−4.91; 0.04	0.053	0.05
External oblique	22.43 ± 15.97	35.31 ± 20.60	−2.88	−26.43; 0.68	0.061	0.11
Lunge Contralateral Load						
Latissimus dorsi	16.27 ± 21.05	20.23 ± 24.64	−3.96	−12.15; 4.23	0.325	0.01
Erector spinae	21.74 ± 22.03	12.26 ± 8.91	9.48	1.26; 17.70	0.026	0.07
Rectus abdominis	10.58 ± 9.41	9.14 ± 8.06	1.45	−3.93; 6.83	0.581	0.01
External oblique	39.19 ± 26.96	22.25 ± 14.32	16.94	4.15; 29.74	0.012	0.14
Lunge Bilateral Load						
Latissimus dorsi	11.44 ± 8.17	16.61 ± 15.88	−5.17	−12.18; 1.84	0.140	0.04
Erector spinae	15.69 ± 15.33	16.01 ± 10.06	−0.32	−3.88; 3.24	0.853	0.00
Rectus abdominis	6.65 ± 3.36	8.62 ± 6.77	−1.97	−4.65; 0.72	0.142	0.03
External oblique	26.10 ± 19.39	21.64 ± 13.73	4.48	−1.57; 10.50	0.139	0.01

Abbreviations: RMS, root mean square; MVIC, maximal voluntary isometric; CI, confidence interval; ŋ^2^, effect size eta square.

**Table 3 healthcare-11-00916-t003:** Differences for sex between lunges according to side and muscle in the different muscles studied.

				Men				Women		
	Lunge vs. Lunge		Mean	95% CI	*p*	ŋ^2^	Mean	95% CI	*p*	ŋ^2^
Anterior Side										
	LIL	LCL	1.30	−2.38; 4.98	1.000	0.01	7.21	−4.60; 19.03	0.379	0.02
Latissimus dorsi	LIL	LBL	1.88	−1.05; 4.81	0.328	0.00	12.05	−2.01; 26.10	0.110	0.08
	LCL	LBL	0.58	−0.98; 2.13	1.000	0.00	4.83	−7.09; 16.75	0.907	0.02
	LIL	LCL	−5.18	−9.19; −1.16	0.009	0.07	−10.22	−19.72; −0.73	0.032	0.09
Erector spinae	LIL	LBL	−1.53	−3.88; 0.83	0.319	0.03	−4.17	−9.70; 1.36	0.189	0.03
	LCL	LBL	3.65	0.92; 6.39	0.007	0.03	6.05	1.84; 10.26	0.004	0.02
	LIL	LCL	−0.55	−2.70; 1.60	1.000	0.00	−3.01	−8.00; 1.99	0.393	0.04
Rectus abdominis	LIL	LBL	0.93	−0.38; 2.24	0.236	0.01	0.92	−0.73; 2.58	0.484	0.01
	LCL	LBL	1.48	0.30; 2.65	0.011	0.04	3.93	−1.23; 9.09	0.181	0.07
	LIL	LCL	−17.47	−29.21; −5.73	0.003	0.18	−16.76	−35.13; 1.62	0.082	0.13
External oblique	LIL	LBL	−0.70	−4.42; 3.02	1.000	0.00	−3.67	−16.22; 8.89	1.000	0.01
	LCL	LBL	16.77	6.69; 26.84	0.001	0.14	13.09	3.96; 22.22	0.004	0.07
Posterior Side										
	LIL	LCL	−2.98	−8.31; 2.35	0.480	0.01	−3.20	−15.31; 8.90	1.000	0.01
Latissimus dorsi	LIL	LBL	0.86	−2.15; 3.87	1.000	0.00	0.42	−7.95; 8.78	1.000	0.00
	LCL	LBL	3.84	−0.15; 7.83	0.062	0.03	3.62	−2.99; 10.23	0.504	0.01
	LIL	LCL	5.88	0.75; 11.02	0.022	0.14	11.63	3.11; 20.15	0.006	0.23
Erector spinae	LIL	LBL	2.17	−2.97; 7.31	0.849	0.01	7.88	1.77; 14.00	0.009	0.11
	LCL	LBL	−3.71	−8.20; 0.77	0.129	0.04	−3.75	−6.95; −0.55	0.019	0.04
	LIL	LCL	3.08	−0.23; 6.40	0.074	0.05	0.87	−3.87; 5.61	1.000	0.00
Rectus abdominis	LIL	LBL	3.07	−0.22; 6.36	0.073	0.05	1.39	−0.45; 3.23	0.186	0.01
	LCL	LBL	−0.02	−0.82; 0.79	1.000	0.00	0.52	−3.78; 4.81	1.000	0.00
	LIL	LCL	11.47	1.45; 21.50	0.022	0.14	13.06	−1.38; 27.51	0.085	0.12
External oblique	LIL	LBL	11.33	2.39; 20.26	0.010	0.14	13.68	3.21; 24.15	0.008	0.13
	LCL	LBL	−0.15	−4.03; 3.74	1.000	0.00	0.61	−7.56; 8.79	1.000	0.00

Abbreviations: CI, confidence interval; LIL, lunge ipsilateral load; LCL, Lunge contralateral load; LBL, lunge bilateral load.

## Data Availability

The data presented in this study are available upon request from the corresponding author.

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
