# Peer review of "Effect of Load Distribution on Trunk Muscle Activity with Lunge Exercises in Amateur Athletes: Cross-Sectional Study"

_healthcare, 2023, doi:10.3390/healthcare11060916_

Round 1

Reviewer 1 Report

The authors wrote an interesting manuscript about the effect of load distribution on trunk muscle activity with lunge exercises. The expertise of the authors is evident from the text.

Introduction

The "Introduction" is relevant to the topic and generally provides a quality background for the research. The study's aim is clear.

Lines 39, 41, 43, 47, 51, 53, 55, 58, 59, 63, 68, 72, 74, 75, 80, 81, 83, 87, 89: Authors are using incorrect citation style. Full-stop should be after the reference number, not before. Unfortunately, this occurs throughout the whole text.

Line 86: "…lunges. .Also…" – Additional full-stop.

Materials and Methods

Part "Materials and Methods" is mostly comprehensible but includes a few shortcomings.

Lines 104 and 105: Please, do not state the average values for the whole sample; these data are not saying anything; they are irrelevant. Use average values separately for females and males. It should be "body height" and "body mass."

Line 213: Full-stop should be after the "(Fig. 1)."

Line 220: Full-stop should be after the "(Fig. 2)."

Line 228: Did the subject on pictures give explicit permission to use the photo for research purposes? Better to use anonymous images – without identification.

Results, Discussion, and Conclusions

Results are presented adequately. The Discussion and Conclusions are conducted within the results' framework. However, I cannot see why the authors presented results for females and males together. From my point of view, this fact is a significant shortcoming of the paper. The authors should differentiate the results according to sex.

Line 244 and Table 1: Details about the research sample should be in the section Materials and Methods. Instead of "Gender," it should be "Sex."

Lines 281, 284, 298: First names of cited authors should not be stated.

Line 304: Should be “Calatayud et al. (49).”

In the part Conclusions authors summarize the main findings. However, practical recommendations are missing – what is the study's outcome? How could the results of this study help us in real life?

References

In-text citations and a list of references are not according to the citation style/standards of the journal!

In conclusion, the manuscript is well-written but includes a few shortcomings. I recommend that the authors divide the research sample into females and males. Therefore, the authors should present the results separately for females and males. Based on this fact, the authors could draw generally valid conclusions. As I mentioned above, the transfer of findings to real life is missing. I consider practical recommendations to be one of the essential parts of the paper. Moreover, the paper includes many mistakes (misspellings, typos, unnecessary commas, and punctuation mistakes).

If the authors are willing to accept the comments and apply them to the text, I recommend the paper for another review.

Author Response

Review 1

The authors wrote an interesting manuscript about the effect of load distribution on trunk muscle activity with lunge exercises. The expertise of the authors is evident from the text.

Introduction

The "Introduction" is relevant to the topic and generally provides a quality background for the research. The study's aim is clear.

Thank you for your comment. We thank the reviewer for his comments and time spent reviewing the manuscript, which we are sure will help us to improve the manuscript.

Lines 39, 41, 43, 47, 51, 53, 55, 58, 59, 63, 68, 72, 74, 75, 80, 81, 83, 87, 89: Authors are using incorrect citation style. Full-stop should be after the reference number, not before. Unfortunately, this occurs throughout the whole text.

Thank you for your comment. The citation style has been modified to the requirements of the journal. In addition, typographical errors have been revised.

Line 86: "…lunges. .Also…" – Additional full-stop.

Thank you for your comment. Done

Materials and Methods

Part "Materials and Methods" is mostly comprehensible but includes a few shortcomings.

Lines 104 and 105: Please, do not state the average values for the whole sample; these data are not saying anything; they are irrelevant. Use average values separately for females and males. It should be "body height" and "body mass."

Thank you for your comment. Modifications of data and text have been made in the manuscript.

Line 213: Full-stop should be after the "(Fig. 1)."

Thank you for your comment. Done

Line 220: Full-stop should be after the "(Fig. 2)."

Thank you for your comment. Done

Line 228: Did the subject on pictures give explicit permission to use the photo for research purposes? Better to use anonymous images – without identification.

Thank you for your comment. The subject has given consent, and a signed copy has already been sent to the journal, reflecting the authorization for publication in the format requested by the journal.

Results, Discussion, and Conclusions

Results are presented adequately. The Discussion and Conclusions are conducted within the results' framework. However, I cannot see why the authors presented results for females and males together. From my point of view, this fact is a significant shortcoming of the paper. The authors should differentiate the results according to sex.

Thank you for your comment. A sex analysis has been carried out, and a table with data broken down by gender has been included.

Line 244 and Table 1: Details about the research sample should be in the section Materials and Methods. Instead of "Gender," it should be "Sex."

Thank you for your comment. Table 1 has been moved to material and methods following their suggestions. In addition, the information has been segmented by sex.

Lines 281, 284, 298: First names of cited authors should not be stated.

Thank you for your comment. These mistakes in the text have been corrected.

Line 304: Should be “Calatayud et al. (49).”

Thank you for your comment. Done.

In the part Conclusions authors summarize the main findings. However, practical recommendations are missing – what is the study's outcome? How could the results of this study help us in real life?

Thank you for your comment. The conclusions have been reformulated to include some practical recommendations.

References

In-text citations and a list of references are not according to the citation style/standards of the journal!

Thank you for your comment. The citation style has been modified to the journal's standards.

In conclusion, the manuscript is well-written but includes a few shortcomings. I recommend that the authors divide the research sample into females and males. Therefore, the authors should present the results separately for females and males. Based on this fact, the authors could draw generally valid conclusions. As I mentioned above, the transfer of findings to real life is missing. I consider practical recommendations to be one of the essential parts of the paper. Moreover, the paper includes many mistakes (misspellings, typos, unnecessary commas, and punctuation mistakes).

If the authors are willing to accept the comments and apply them to the text, I recommend the paper for another review

Again, we thank the reviewer for his comments and suggestions. All modifications proposed by the reviewer have been implemented.

Reviewer 2 Report

This study provides interesting information on trunk muscle activity associated with load distribution during lunge exercise. While the findings are worthy of publication in Healthcare, some changes and clarifications regarding the methods and results should be made.

1. Did the participants experience muscle fatigue in the context of 1-minute rest and how does that affect measurement data?

2. More details of the linear mixed model are required, e.g, is it a multivariate mixed model? What are the fixed and random effects factors? Does the model include other confounding variables such as age and sex?

3. Table 3 is missing. Please also provide more explanation and analysis of interaction between lunge type and loading side for four trunk muscles.

4. Minor typos (e.g., line 35, line 64, line 144, etc.) and duplicate sentences (line 232) need correction.

Author Response

Review 2

This study provides interesting information on trunk muscle activity associated with load distribution during lunge exercise. While the findings are worthy of publication in Healthcare, some changes and clarifications regarding the methods and results should be made.

  1. Did the participants experience muscle fatigue in the context of 1-minute rest and how does that affect measurement data?

Thank you for your comment. The sample was composed of young athletes. They did not experience fatigue during exercise or at rest. Also, to avoid this possible effect on the results, the order of exercise execution was randomized for this purpose.

  1. More details of the linear mixed model are required, e.g, is it a multivariate mixed model? What are the fixed and random effects factors? Does the model include other confounding variables such as age and sex?

Thank you for your comment. The linear mixed model used is a multivariate mixed model. This linear model used a fixed factor (between-subject “anterior-posterior” and a repeated measures factor “type of exercise”). In our case 2X3.

In this 2X3 one-way mixed ANOVA model the SPSS program allows to modify the SPSS syntax and to obtain the post-hoc data. These ANOVA models use regression models, so the covariance is constant.

A new statistic has been added with sex as a covariable.

We have modified the text of the manuscript to improve the explanation.

  1. Table 3 is missing. Please also provide more explanation and analysis of interaction between lunge type and loading side for four trunk muscles.

Thank you for your comment. We apologize for this mistake. It is a typographical mistake.

This analysis has been carried out and a new table has been added to the text, as suggested by the reviewer.

  1. Minor typos (e.g., line 35, line 64, line 144, etc.) and duplicate sentences (line 232) need correction.

Thank you for your comment. The spelling mistakes and the duplicate sentences have been corrected.

Round 2

Reviewer 1 Report

I want to thank the authors; they improved the paper according to the provided comments.

Table 1 is mentioned twice in the text (line 113 and line 241). Please, remove Table 1 from line 241 – part Results.

Line 141: The word “maesure” – misspelling.

Lines 385–387: The title of reference 15 should not be capitalized.

Lines 469–471: Same as the previous comment. The title of reference 50 should not be capitalized.

Based on the comments, the authors significantly approved the manuscript. They provided extended changes mainly concerning the division of the research sample and data analysis. If the authors make the final changes, I can recommend publishing it in Healthcare.

Author Response

I want to thank the authors; they improved the paper according to the provided comments.

We thank the reviewer for his time and dedication to improving the article.

Table 1 is mentioned twice in the text (line 113 and line 241). Please, remove Table 1 from line 241 – part Results.

Sorry for the repetition mistake. The table has been deleted.

Line 141: The word “maesure” – misspelling.

Thank you for your correction. The mistake has been corrected.

Lines 385–387: The title of reference 15 should not be capitalized.

Thank you. The reference has been corrected.

Lines 469–471: Same as the previous comment. The title of reference 50 should not be capitalized.

Thank you. The reference has been corrected.

Based on the comments, the authors significantly approved the manuscript. They provided extended changes mainly concerning the division of the research sample and data analysis. If the authors make the final changes, I can recommend publishing it in Healthcare.

Thank you again for your contributions to the improvement of the manuscript. 

Reviewer 2 Report

The authors have addressed my concerns and I recommend acceptance of the manuscript.

Author Response

Thank you for your comments and your time in improving the manuscript.

Round 3

Reviewer 1 Report

The authors modified the text according to the comments and therefore I recommend publishing the article in Healthcare.

Author Response

(The authors gave the same response as above.)
